# The Impact of Marathons on the Recovery of Heart Rate and Blood Pressure in Non-Professional Male Marathoners’ (≥45 Years)

**DOI:** 10.3390/medicina57121346

**Published:** 2021-12-09

**Authors:** Ülle Parm, Anna-Liisa Tamm, Andras Laugamets, Margus Viigimaa

**Affiliations:** 1Physiotherapy and Environmental Health Department, Tartu Health Care College, Nooruse 5, 50411 Tartu, Estonia; ylleparm@nooruse.ee; 2The Tartu Ambulance Foundation, Riia 18, 51010 Tartu, Estonia; kiirabi@kiirabi.ee; 3Department of Health Technologies, Tallinn University of Technology, Ehitajate tee 5, 19086 Tallinn, Estonia; margus.viigimaa@taltech.ee; 4The North Estonia Medical Centre, J. Sütiste tee 19, 13419 Tallinn, Estonia

**Keywords:** non-professional marathoners, body composition, blood pressure, heart rate, nutrition

## Abstract

*Background and Objectives:* Physical activity has a positive impact on health, and the participation in exercise and sports, including marathons, has increased in popularity. This kind of sport requires extreme endurance, which can cause different health problems and even lead to death. Participants without sufficient preparation and, in particular, men 45 years of age and older belong to a high risk group. The aim of this study was to determine the impact of marathons and cofactors associated with marathons on the recovery of heart rate (HR) and blood pressure (BP) of non-professional ≥ 45 years old male marathoners. *Materials and*
*Methods*: A total of 136 ≥ 45 year old, non-professional (amateur marathoner), male participants were recruited. Data collection involved a questionnaire, body composition measures, and BP and HR results before and after finishing the marathon. Descriptive data, *t*-test, Mann–Whitney or χ^2^ test, and Pearson’s correlation were applied. *Results*: Participants (skiing *n* = 81, cycling *n* = 29, running *n* = 26; mean age 51.7 ± 7.1 years old) had previously attended a median of 35 (IQR 17.5–66) marathons and travelled 2111.5 (IQR 920–4565) km. Recovery of HR and BP after finishing and recovery time was insufficient and not associated with marathon preparation. Running was the most burdensome for HR, and cycling was most taxing for BP. Chronic diseases did not influence participation in the marathon. *Conclusions*: The preparation for the marathon was mainly sufficient, but recovery after the marathon was worrisome. Marathons are demanding for ≥45 year old males and may be too strenuous an activity that has deleterious effects on health.

## 1. Introduction

Regular physical activity (PA) has been praised as an essential part of a lifestyle in the promotion of cardiovascular (CV) health [1,2,3]. Due to this knowledge, during recent decades, the number of participants in marathons has significantly increased and more professional and recreational marathoners can be seen participating in this endurance sport that requires excessive exertion [4,5].

Vuori et al. [2] compared exercise with medicine, and believes that it should be dealt with in the same ways as pharmaceuticals and other medical interventions, for instance, each drug has under and overdoses. The previously shown relationship between a person’s PA level and heart health led to the conclusion that “if some exercise is good, more must be better” [6], but high levels, as with lower doses of exercise, may both be associated with CV health [6,7]. The prevalence of sudden cardiac death for marathon runners is 1/50–200,000/year [1,4,8]. Unfortunately, a specific dose (extent and duration) of exercise that reduces or leads to CV disease risk and mortality is still unclear [7].

The age of participants is increasing, and especially men in ages 40–60 prefer longer distances [5]. So, among the marathoners, many are middle-aged persons, usually non-professionals, who are playing sports in their free time [8]. Unfortunately, in middle-aged males who have taken part in marathons since the year 2000, the frequency of cardiac arrests and sudden death has increased [9,10]. As a professional marathoner has a training team, nonprofessional, recreational, amateur, or leisure athletes are individuals (usually ≥35 years old) who engage in sports on regular or an inconsistent basis, participate in different informal sport competitions, usually not requiring systematic training or the pursuit of excellence [9]. Therefore, their training status is not commonly controlled. Participants that are men without sufficient preparation and over 45 years old belong to a risk group [9,11]. However, those who take marathons seriously must prepare for the activity [8].

Regular, voluminous, well-advised training is the key to successfully completing a marathon [12,13], which will provide physical readiness for the marathon and helps to prevent serious health problems and injuries [14,15]. Professional athletes usually train at high volumes according to specific training plans. Yet, we do not know if the preparation of recreational marathoners is sufficient. Moreover, food and fluid consumption (daily, pre-marathon and during the marathon) plays a role in sports performance and maintaining good health [16]. There is no specific guidance for eating habits, fluid, and supplement and vitamin consumption of Estonia’s non-professional male marathoners. It is possible that marathoners will participate in a competition without proper preparation, risking their own health.

There are limited studies that have looked at the impact of marathons and co-factors on nonprofessional marathoners’ heart rate (HR) and blood pressure (BP) recovery. Limited information is available for recreational marathoners, although relevant data is needed to develop consensus-based guidelines regarding athletes in special populations [1]. The aim of this study was to describe the impact of a marathon on HR and BP recovery of selected physiological parameters in non-professional male athletes aged above 45 years old (who are at risk based on the literature).

## 2. Materials and Methods

The sample consisted of ≥45 year-old non-professional, Tartu Marathon Cube (consists of six events) [17] registered men who agreed to participate in the study (*n* = 179). They had to have registered for and finished one or more of three marathon stages (skiing 17 February 2019; cycling 26 May 2019, running 05 October 2019). The environmental conditions such as temperature (°C), humidity (%) and average wind speed (m/s) were 2.4/65/2.8; 15.9/80.1/2.6; and 18.1/77.3/3.0, respectively [18]. Firstly, 24–48 h prior to the stage, the study participants filled out a questionnaire about their age, incidence of previously diagnosed chronic disease, used medications, previous participation in different marathons and passed kilometers, preparation for marathon (hours and times per week) during the prior 3 months, and their eating habits. The questionnaire was composed by the authors of the article and tested (pilot study) using a representative sample including eight physically active adult men. As a result of the pilot study, the description of sports equipment and training conditions in the questionnaire was shortened; the terms used in the questionnaire were adjusted to make them more comprehensible to the respondent.

In the same day (24–48 h prior to the marathon), their body composition, BP, and HR in rest condition were measured. BP was measured (systolic, diastolic) by using a sphygmomanometer on both arms. The participants had to sit down and relax for about 3 min before BP measurement. The upper arm being used was rested on a table, at about the same height as the heart, while the reading was being carried out. The body composition (fat %, fat free mass, total body water) was determined with a Seca mBCA 525 medical body composition analyzer (Hans E. Rüth, Hamburg, Germany) according to the manufacturer’s instructions [19].

The body mass index (BMI) calculation formula is body mass (kg) divided by height squared (m^2^). Body weight is classified on the basis of BMI from the age of 20 as follows—underweight (index value < 18.5); normal body weight (18.5–24.9); predominance (25.0–29.9); stage I obesity (30.0–34.9); stage II obesity (35.0–39.9); and stage III obesity (>40) [20]. The healthy fat percentage range for men in the 41–60 age group is 11–22%. In overweight, the fat percentage is in the range of 22–27% and in obesity is >27% [21].

After finishing the marathon, participants were unable to sit down immediately and moved to an investigators room at an agreed location, which was situated not more than 5 min from the finish line. There, as soon as the marathoner uncovered his right hand, his BP and HR were measured (about 5 min after finishing). After, they had to relax and answer to investigators’ questions (ill-being, circumstance on racecourse, water consumption, visiting the K-minus). After 10 min of rest (15 min after finishing), the measures of BP and HR were repeated [22]. In further analyses, only leading hand systolic parameter was used. Forty-three participants were excluded because some of them did not start, did not finish, or did not find investigators (Figure 1).

The software programs Sigma Plot for Windows version 11.0 (GmbH Formation, Berlin, Germany) and R 2.6.2 were used for statistical analysis. Differences in proportions were compared with Chi-square test or Fisher exact test; in continuous variables by Student’s *t*-test or Mann–Whitney test, and data before and after the marathon in the same person by paired sample *t*-test, as appropriate. To assess the impact of different factors (passed whole or half marathons): skiing (63 or 31 km), cycling (124 or 56 km), running (42 or 21 km); age, weight, BMI, incidence of chronic disease (separately CV); previous participation in different marathons and passed km-s; preparation for the marathon (hours and times per week) over 3 months; consumption of liquids 1 day prior and during the marathon; fat %; fat-free mass, whole and extracellular liquid on HR (ΔHR5 change between 5 min after finishing and rest HR) and BP (ΔBP5); changes during relax time (ΔHR or ΔBP—change between 15 min after finishing and results 5 min after finishing); and recovery after finishing (ΔHR15 or ΔBP15—change between results 15 min after finishing and rest results). Univariate linear regression analysis was applied. A significant difference was accepted when *p* < 0.05. For calculating correlations, the Spearman test was used.

## 3. Results

### 3.1. Demographic Data, Prior Marathon Experience and 3-Month Preparation

Participants (aged 51.7 ± 7.1 years), although nonprofessional, had substantial marathon experience. The median of participation in previous marathons was 35 and passed 2111.5 km. Prior to each marathon, subjects trained on mean 4.8 times and 7.2 h per week (Figure 2A,B). The training prior to the skiing marathons involved more hours but less time per week compared to running and cycling marathons (Table 1). Overall, 98.5% of the participants had taken part in at least one marathon previously. Three of the participants participated in both the ski and cycling marathon, two in both the cycling and running marathon, and one in all three stages. The preparation style was proportional with participators in different marathons.

### 3.2. Body Composition

The results demonstrate that runners, compared to cross-country skiers, had lower body fat mass (10.8 kg vs. 14.3 kg; *p* = 0.01) and fat percentage. Runners also had lower body fat mass (10.8 kg vs. 15.7 kg; *p* = 0.016) and fat percentage than cyclists. Runners had lower body mass than cyclists and lower BMI than skiers. Previously completed marathons or a training load of 3 months prior to the marathon showed no association with body composition parameters. Previously completed marathons and their passed distance were correlated with 3 months of prior marathon training times (r = 0.93; *p* < 0.001 and r = 0.90; *p* < 0.001, respectively) and training hours (r = 0.92; *p* < 0.001 and r = 0.95; *p* < 0.001, respectively) in the week. Body composition had no correlation with cross-country skiing marathon performance. Total body water content was similar in all subjects regardless of which marathon they participated in, but there was a positive correlation with waist circumstance (r = 0.41; *p* < 0.001), body mass (r = 0.73; *p* < 0.001), and BMI (r = 0.48; *p* < 0.001) similarly with extracellular water.

There were no correlations between body composition data and speed with distance marathoners passed. We monitored this in full-marathon skiers (*n* = 73) only, as in other distances, the participation was low. The average speed was 4 h 25 min ± 47 min (fastest 3 h 1 min, slower 6 h 34 min).

### 3.3. Health Condition and Medication

Thirty six percent of marathoners were diagnosed with some form of chronic disease. The most common diagnoses were CV diseases (11.8%) and sport traumas (8.8%). Relax time BP ≥ 140 mmHg occurred in 27% of participants, and the higher assessed level was 190 mmHg. A total of 52.9% of the participants had used any medication within the last year (<10 days to >180 days), most commonly those controlling BP and other CV issues. Irregular painkillers (42.6%) were used, including (6%) before or during the marathon. Seventy-six percent assessed their sleep quality as good. Only 27.9% regularly completed an annual health check-up; 45.6% of them within the last year. There were no statistical differences between previous marathon participants or those who prepared for 3 months prior to the marathon when comparing those without any chronic diseases. Some subjects had high BP values (systolic > 200 mmHg) and also had arrhythmias (based on their HR), so the researchers did not recommend for them to participate in the marathon.

About half of the marathoners (41.9%) reported some health complaint during the marathon, including more frequent pain, aches, and discomfort in legs (13.7%; skiers 12%, runners 7.7%, cyclists 23.3%); and skeletal muscle cramping (9.4%; skiers 9.6%, runners 19.2%). The cyclists felt pain in the back more often (16.7%), and one of them needed medical support (pain in the back and left knee). Injuries due to falls were only reported in skiers (8.4%).

### 3.4. Water Consumption Prior and during Marathon

All marathoners were omnivores. A total of 76.4% of the marathoners monitored their regularity of eating daily and 72.8% while preparing for the marathon. The day before the marathon, the fluid consumption was 1.9 L (skiers, cyclist, runners, full and half marathoners: 1.95; 2.0; 1.7; 2.0; and 1.7, respectively), and water (59.6%), mineral water (25.8%), juice (21.3%), and sports drinks (18.4%) were preferred. During the marathon, the fluid consumption was 0.5 L/h, and the favorite liquids were sports drinks (72.8%) and water (51.5%) (Table 1). Only 14.8%, 9.9%, and 4.9% of skiers used tea, bouillon, and coffee, respectively. The fluid consumption during the marathon was not correlated with total body and extracellular water recon with all participants and different marathons, but the day before, the marathon consumption was positively correlated with extracellular water in the whole study group (r = 0.25; *p* = 0.006).

### 3.5. HR and BP after Marathon and Recovery

Preparation before the marathon was not associated with HR and BP. Table 2 shows results of HR and BP prior to the marathon, 5 (measured immediately after finishing) and 15 min after finishing (10 minutes’ recovery time). After finishing in all distances, HR was in general >20 beats faster and differed significantly from the rest time parameters. During the relax time, the HR was not reduced in comparison with rest time results as expected, but differed also from results measured after the relax time. However, the HR recovery after 10 min rest did not happen, as results measured after recovery differed significantly from rest time parameters.

The rest time and after finishing BP parameters did not differ in runners and those who passed the half-distance. In general, there were 65 marathoners whose BP after finishing were >5 mmHg lower (max −55 mmHg) and 33 (skiers *n* = 7, cyclists *n* = 13, runners *n* = 12; max 55) whose BPs were >5 mmHg higher compared with rest time parameters. Although parameters results differed between the finish and after relax time (except skiers), in all distances the parameters after relax time were statistically lower than rest time. During relax time in 95 (69.9%) of participants, the results of BP were fallen (64.2%, 100% and 92.3% from skiers, cyclists and runners, respectively; max −55 mmHg), and in 25 (18.4%) were raised (29.6% and 4.2% from skiers and runners, respectively; max 22 mmHg). From all participants, 91 (66.9%) had BPs >5 mmHg after relax time, with lower results (mean −18.7 ± 9.3; max −65) than rest time.

To clarify the reason for the flow-off of BP after relax time, for 3 months in two groups (BP after finishing ≥6 mmHg lower vs. others), different parameters (age, previous participation in different marathons and passed km, preparation for marathon (hours and times per week)) were compared. The only parameter that differed was in the group with better recovery and previously passed a longer distance of km (in median of 2499.5; IQR 1013–4680 vs. 1695; IQR 749–3310.8; *p* = 0.024).

### 3.6. Factors Influencing HR and BP

The results of univariate linear regression analyses data presented the influence of different factors to BP and HR after finishing the marathon, 10 min after recovery time. Changes during relax time are presented in Table 3.

Running was the most burdensome to HR, and cycling was taxing to BP. Results of univariate logistic regression analysis showed that among monitored factors, only weight and fat-free mass positively influenced HR after finishing in comparison with rest rime. For example, calculating prognostic HR in marathoners with 60, 70, and 80 kg fat-free mass, the rise of HR after marathon would be 21.4, 25.6, and 29.8 HR, respectively. The changes in BP after finishing in comparison with rest time were higher in skiers in comparison with cyclists and runners, and also those who passed a long distance in comparison with a half marathon.

During the relax time, the HR in younger participants recovered better. For example, calculating prognostic HR change in 45, 55, and 65 year-old marathoners, the fall of HR after relax time would be 10.2, 6.8 and 3.4 beats, respectively. In this period, the further flow off of BP was smaller in skiers than cyclists and runners and those who passed whole vs. half distance. In subjects with a higher fat-free mass and a greater amount of whole body and extracellular liquid 1–2 days prior, the changes during recovery time were smaller.

After relax time, in comparison with rest time, the recovery of HR was supposed and none of the monitored factors influenced the recovery of this parameter. Monitoring BP recovery, in skiers’ vs. runners and full vs. half marathoners, the differences in comparison with the rest time remained. Differently with changes during relax time, the subjects with a higher fat-free mass and higher amount of whole body and extracellular liquid, the results after the recovery time were better.

## 4. Discussion

In planning this study, we assumed that study participants were less-experienced nonprofessional marathoners, and we planned to assess their arrangements for participation in marathons and their 10 min BP and HR recovery after the marathon. A recreational marathoner is deemed an athlete, who usually is middle-aged or elder, who does not require systematic training (without a trainer), and participates in different informal recreational sports either regularly or unsystematically [9]. The participants of this study, although non-professionals, had substantial marathon experience, and for some of them, the marathon is their style of life [23]. Our study participants’ preparation for the marathon was mainly sufficient, and most of the participants (98.5%) had taken part in at least one marathon beforehand. Our study showed that recovery was dependent on the type of marathon (cross-country skiing, cycling, running), distance (full vs. half marathon), previously passed kilometers, body water content, and fat-free mass.

Our results showed that running was the most burdensome to HR, and cycling was the most taxing to BP. Moreover, we confirmed that intensive training slows the resting HR [24,25]. Usually, exercise increases cardiac output and BP [7]. The acceleration of HR during and after exercise is normal, as increasing oxygen demand from the body is compensated by increased cardiac output and HR [26]. The higher weight related with bigger muscle mass (fat-free mass) had a positive impact on HR after finishing in comparison with rest time. The results can explain that runners, compared to cross-country skiers and cyclists, had lower body fat and fat percentage. On the other hand, runners had lower body mass than cyclists and lower BMI than skiers. BP is dependent on peripheral vascular resistance and cardiac output [26]. The results after finishing were lower in comparison with rest time results in skiers and those who had completed a full marathon, and higher (not significant) in cyclists, runners and also those who had completed half-distance marathons. It has been previously described that after excessive training, systolic BP usually declines because of the rapid decrease in cardiac output [26]. Therefore, we can conclude that exertion during skiing is harder.

We also monitored recovery of HR and BP during 10 min relaxing time and after relaxing time in comparison with rest time results, as well as factors that influence these differences. The changes of HR and BP (excluding skiers) in all distances during relax time were significant, and the recovery was not sufficient. In a study carried out with <65 year-old ultra-marathon runners, the expected increase of HR from baseline during the race, which later decreased during recovery, was after 60–90 min, still higher than resting levels [27]. Moreover, it has been shown that after a skiing marathon the cardiac vagal activity reduced for at least 5 h, while recovery occurred no later than 30 h after [28]. The expected recovery could have been too fast, although 10−15 minutes were used for recovery reference by others [22,25]. Different marathons and distances influence recovery differently, especially BP. However, the HR showed the trend to recovery, but parameters stayed after 10 min recovery in comparison with rest results, showing a statistically different value.

Despite most marathoners being experienced, the recovery of HR and especially BP was not sufficient. BP after recovery time in all marathons and distances was lower than in rest time. This is normal, as after maximum exercise, BP can reach resting levels or lower within 6 minutes, and may remain lower than rest levels for several hours [26]. The similar results by Taksaudom et al. [22] in Thailand showed where the BP and HR stayed similarly different from rest results after post-marathon 15 min recovery (135.9 mmHg vs. 119.7 mmHg and 72.6 vs. 96.4, respectively; *p* < 0.001). Regardless, for some marathoners, the decrease of BP was significant (−65 mmHg), which might lead to exercise-associated collapse. It is explained by the result of lower extremity pooling of blood in a time when the athlete stops exertion and impairment of cardiac baroreflexes [28], by dehydration and heat loss to the environment during cold weather [29]. This did not happen with participants during the study (the weather was not cold in every race), but some of the participants did not feel well themselves.

The exact HR and BP recovery influencers during 10 min relaxing time and recovery are not easy to explain. For example, there was a negative correlation between HR recovery during recovery time, meaning that during this time, the changes in HR value in older people were smaller. On the other hand, the results after recovery in older people were more similar with rest results than in younger participants. We can presume that elders passed the distance more wholesomely, and their HR was not so high after finishing (not in this level that manifested as an influencing factor). It has previously been shown that younger runners showed the fastest end spurts compared with older marathoners [30]. However, this result is not usual, as it has previously been shown that normal systolic BP response to progressive exercise is dependent on age and is higher with advancing age [26]. Thus, marathon as a lifestyle is associated with specific behaviour factors such as being engaged in regular training for a few years [12]. Therefore, master endurance marathoners in older age groups might show similar performance aspects as younger marathoners due to their previous exercise experience [31].

During the 10 minutes, changes of BP parameters were smaller in skiers (the change was not significant) in comparison with runners and cyclists, and in full marathon participants in comparison with half-marathon participants. However, in comparing the results after relax time with rest time, the bigger changes of BP stay in skiers in comparison with runners and those that had full distance in comparison to half distance. Changes were smaller in participants with higher weight, BMI, and fat-free mass (it means persons with higher skeletal muscle mass), and those who had a larger amount of whole and extracellular body water. In general, comparing results after relax time, we can say that recovery was better in those whose fat-free mass, whole body, and extracellular water was greater. Thus, for endurance marathoners attempting to minimize dehydration [16], but also avoid over-hydration, prior marathon is important. On the other hand, the BP changes during recovery time were greater in those who had previously passed more km in marathons. However, if their BP parameters were monitored after finishing in those who had passed more km previously in comparison with those who had passed less km-s, their summarized falling was smaller (−145 vs. −379 mmHg).

More than one third of marathoners were diagnosed with chronic diseases, but they did not influence their participation in the marathon. The most common were CV diseases and sport traumas. This number is quite similar to Hollander et al. [32], who found that 37.3% (*n* = 3213) of recreational runners had health problems during preparation for a marathon, although in this study, acute problems (injuries, illness) were monitored [32]. The complaints during the marathon, in less than half of marathoners (pain and muscle cramps), were similar to those reported by others [33], although our participants did not complain of heat exhaustion due to climate conditions. More than half had used medication within the last year (<10 days to >180 days), most commonly those controlling BP and other CV issues. Painkillers were used irregularly, including before or during the marathon. The usage of medication was rather common. Ibidem, non-steroidal inflammatory medications (usually used as painkillers) have been implicated as a risk factor for development of hyponatremia [34].

Pre-race preparation needs special training as well as optimal food and water consumption [16] that depend on factors such as distance duration [13,35], the type of marathon, and climate conditions. Traditionally preparing for marathons involves a high training volume and long endurance runs, but it is a challenge to find a training volume that is enough for optimal performance. For example, preparing for full marathons should involve runs at least 8 (5–10) km several times (3–5 days) a week for 16–20 weeks [36]. Training for a half marathon should begin 2–6 weeks before a running event; the training volume > 32 km/week was associated with a faster finish time [37]. However, running > 65 km/week for men has been found to be related to a higher risk of running-related injuries in recreational runners [14]. Unfortunately, we did not monitor a training pace and passed km in training time of our participants, but we presume that preparation was good. Recovery of HR and BP after finishing was insufficient and was not associated with marathon preparation. It has previously been shown that fitness improvement might not always be associated with increased HR recovery [25].

Correlation between different body composition (body mass, fat %, fat-free mass) parameters and race performance has been proved by previous studies [24]. Male half-marathoners are heavier, they have thicker upper arms and thighs, a higher body fat %, and a higher skeletal muscle mass than full-marathoners; this is presumably caused by their fewer years of experience and fewer weekly running hours and travelled km [38]. In our study, previously completed marathons or a training load 3 months prior to a marathon showed no association with body composition parameters. It can be explained by the similar study group who frequently trained and participated in marathons. Similarly, body composition had no correlation with cross-country skiing marathon performance. However, the higher fat-free mass affected the recovery of BP during the relaxation period. According to BMI and fat percentage, most marathoners´ body compositions were healthy.

Total body water content prior to the race was similar in all our marathoners, but extracellular water amount was higher in skiers than runners. Our study showed higher extracellular water results in those whose consumption prior to the marathon was greater, but there were no differences in skiers and runners. Smaller extracellular water was a significant factor in better recovery of BP after relax time, probably due to the fluid over-consumption that is recommended the day before the marathon [16,39]. Thus, it is important to be adequately hydrated before and during the race [40,41,42]. It is recommended that runners drink in such an amount (400–1200 mL/h—our participants were within the recommended range) to prevent excessive dehydration and limit body mass losses through sweating to 2% of body mass [16,43]. Nicolaidis et al. [41] does not suggest introducing more than 300–600 mL/h. There is considerable variability in sweating rates between individuals, so individualized fluid replacement programs are recommended [44].

There are no previous studies where marathoners in Estonia were investigated. This study is one of the few studies focused on the influence of marathon trail running using historical, basic demographic and body composition data, and where finishing data (HR, BP and short questionnaire) quite immediately after the marathon is used. It was complicated due to noise disruption and providing marathoners with privacy upon finishing. Therefore, some part of examinations beside the track were carried out in a limited manner (e.g., [22,45,46]), as with others carried out in hot climate areas, such as Spain, Thailand, and Korea. There are some limitations to this study: (1) The examination was performed at the finish line. This meant we had to find quieter rooms nearby (not more than 5 min). We lost many participants in the sky marathon, as the climate changes in the winter and the finish line changed every day; thus, participants could not find the investigators. (2) The participants who arrived for the study were not quite as expected. The so-called amateur athletes were expected, but they were still marathoners who had years of experience participating in marathons. Only a few participated in the respective marathon for the first time. (3) It is important to evaluate body composition before and after the marathon in order to compare the acute effects of different types of marathons on the body and to assess the fluid consumption of marathoners on the track. Immediately after passing the marathon stage, it is difficult to organize measurements in winter conditions because the degree of fatigue of marathoners and the desire to go home for rest quickly must be taken into account. (4) Due to the already large and burdensome design of the study, other important aspects affecting physical performance (such as sleep quality) were not studied.

## 5. Conclusions

The benefits of physical activity (PA) are well-known, but excessive exercising is not always beneficial for over 45 year-old non-professional marathoners. Even though they were experienced marathoners, their recovery was not satisfactory. Thus, instead of competing with oneself and others, a marathon event at a certain age could be considered a so-called national sports event, and therefore half-marathons could be practiced. Our study showed that running was the most burdensome to HR and cycling was taxing to BP, and for adequate recovery, adequate hydration is needed. It can be agreed that more endurance training may not always be better and that optimal dosing of training regimens and trace frequency are needed [1].

## Figures and Tables

**Figure 1 medicina-57-01346-f001:**
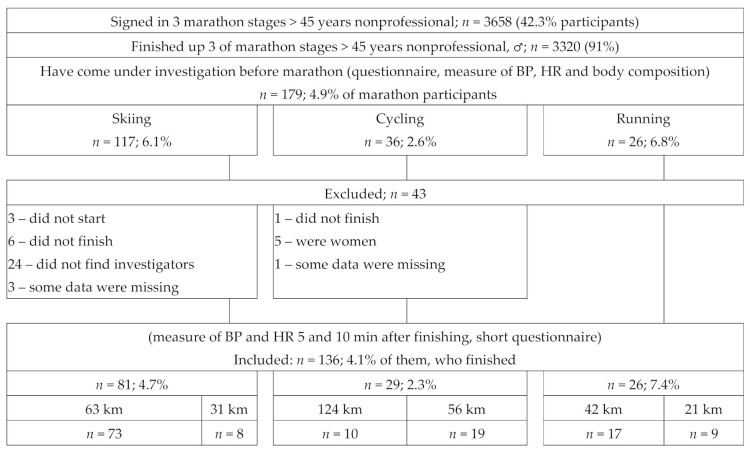
Study design.

**Figure 2 medicina-57-01346-f002:**
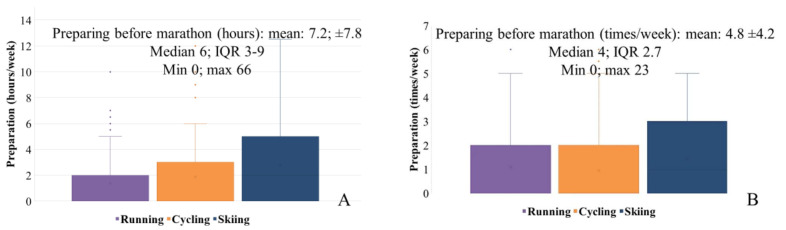
Three mean activities during the 3-month preparation for the marathon (hours (**A**) and times (**B**) are presented for preparing for marathon-independent activities, fitness studios, gymnasium, swimming, and garden tools are included).

**Table 1 medicina-57-01346-t001:** Demographic data and body composition in nonprofessional ≥ 45-year-old male marathoners.

Parameters		All Participants	Skiing	Cycling	Running	Long-Distance	Half Marathon
*n*=		136	81	29	26	100	36
Age (years); mean (SD)		51.7 (7.1)	52.9 (7.6)	50.7 (7.7)	49.4 (3.5)	51.8 (6.8)	51.5 (8.0)
Family; %	Married	66.9	71.6	58.6	61.5	71	55.6
	Cohabitation	18.4	18.5	17.2	19.2	19	16.7
	Single	6.6	3.7	20.7	0	4	13.9
Education; %	Basic	1.5	2.5	0	0	2	0
	Secondary	31.6	16.0	31	38.5	29	38.9
	Higher	66.2	66.7	69.0	61.5	68	61.1
With chronic diseases; %		49.3	43.2	75.9	38.5	41	72.2
With CVD diseases; %		11.8	13.6	6.9	11.5	9	19.4
Training times/week; median (IQR)		4 (2–7)	3 (0.75–6) ^hi^	5 (3–8.6) ^h^	5 (3–7) ^i^	4 (1–7.3)	4 (3–5.8)
Training h/week; median (IQR)		6 (3–9)	11.5 (5–25.5) ^jk^	7 (4.3–10.1) ^j^	5.8 (4.5–10) ^k^	6 (2–9)	5.5 (4–10)
Body mass (kg); mean (SD)		81.5 (9.6)	82.9 (10.3) ^a^	81.7 (8.9) ^b^	77.2 (6.8) ^ab^	80.6 (8.6)	84.2 (11.8)
BMI (kg/m^2^); mean (SD)		25.1 (2.6)	25.5 (2.6) ^c^	25.3 (2.9)	23.8 (1.9) ^c^	24.9 (2.3)	25.9 (3.3)
Fat-free mass (kg); mean (SD)		67.7 (6.3)	68.6 (6.4)	66.5 (6.3)	66,4 (5.7)	67.6 (5.7)	68.2 (7.8)
Fat %; mean (SD)		16.6 (6.2)	16.8 (5.6) ^e^	18.8 (6.6) ^d^	13.7 (6.6) ^de^	15.9 (5.8) ^f^	18.5 (6.9) ^f^
Waist circumference (m); mean (SD)		0.9 (1.1)	0.9 (0.07)	0.9 (0.07)	0.9 (0.09)	0.9 (0.01)	0.9 (0.1)
Total body water (L); mean (SD)		49.7 (4.7)	50.3 (4.8)	48.9 (4.8)	48.7 (4.2)	49.6 (4.3)	49.9 (5.8)
Extracellular water (L); mean (SD)		20.5 (2.1)	20.8 (2.1) ^g^	20.3 (2.0)	19.7 (1.9) ^g^	20.5 (1.9)	20.6 (2.5)
Water prior marathon (L); mean (SD)		1.9 (0.8)	2.0 (0.8)	2.0 (0.9)	1.7 (0.8)	2.0 (0.8)	1.7 (0.8)
Water during marathon (L/h)		0.53 (0.4)	0.47 (0.34) ^l^	0.68 (0.1) ^l^	0.53 (0.01)	0.47 (0.28)	0.69 (0.59)

Training—means 3 months’ prior stage. Statistical differences: ^a^—*p* = 0.018; ^b^—*p* = 0.042; ^c^—*p* = 0.002; ^d^—*p* = 0.023; ^e^—*p* = 0.019; ^f^—*p* = 0.027; ^g^—*p* = 0.022; ^h^—*p* = 0.013; ^i^—*p* = 0.01; ^j^—*p* = 0.014; ^k^—*p* = 0.01; ^l^—*p* = 0.025.

**Table 2 medicina-57-01346-t002:** Results of HR and BP prior to the marathon, 5 and 15 min after finishing.

Marathon		HR (Mean; SD)			*p*=		
	*n*=	Rest Time	5 min	15 min	Rest vs. 5 min	5 min vs. 15 min	Rest vs. 15 min
All participants	136	61.6; 8.8	85.6; 15.8	78.1; 12.8	<0.001	<0.001	<0.001
Skiing	81	61.1; 9.1	84.3; 13.4	76.9; 10.6	<0.001	<0.001	<0.001
Cycling	29	63.3; 9.7	84.8; 16.7	78.8; 12.1	<0.001	0.014	<0.001
Running	26	61.1; 6.5	90.5; 20.8	79.6; 18.9	<0.001	<0.001	<0.001
Long-distance	100	61.0; 8.2	84.7; 15.3	77.1; 13.1	<0.001	<0.001	<0.001
Half-marathon	36	63.1; 10.2	88.1; 17.1	80.7; 12.0	<0.001	0.02	<0.001
		**Systolic BP (Mean; SD)**					
All participants	136	133.2; 14.0	128.3; 17.5	121.2; 11.1	0.001	<0.001	<0.001
Skiing	81	134.4; 13.1	122.5; 13.0	121.1; 10.7	<0.001	0.196	<0.001
Cycling	29	131.1; 16.3	138.7; 20.1	122.7; 13.2	0.012	<0.001	<0.001
Running	26	131.5; 14.8	134.8; 19.6	119.9; 10.1	0.499	<0.001	<0.001
Long-distance	100	133.0; 13.3	124.9; 14.9	119.7; 10.1	<0.001	<0.001	<0.001
Half-marathon	36	133.5; 16.2	137.8; 20.7	125.5; 12.8	0.193	<0.001	<0.001

Paired *t*-test. 5 min—parameters after finishing; 15—parameters after 10 min relaxing.

**Table 3 medicina-57-01346-t003:** Parameters influencing the BP and HR parameters after finishing, during relax time, and recovery after relax time (results of linear regression analysis).

Parameter	Influencing Factor	Coef.	*p*=
ΔHR5 (5-0)	Weight	0.42	0.05
	Fat-free mass	0.30	0.034
ΔBP5 (5-0)	Skiing vs. cycling	−23.95	<0.001
	Skiing vs. running	−15.18	<0.001
	Full vs. half-marathon	−16.06	<0.001
ΔHR (5-15)	Age	−0.35	0.03
ΔBP (5-15)	Skiing vs. cycling	−14.63	<0.001
	Skiing vs. running	−13.59	<0.001
	Full vs. half-marathon	−7.1	0.005
	Previous passed km on marathons	0.04	0.01
	Weight	−0.24	0.045
	BMI	−0.87	0.047
	Fat-free mass	−0.18	0.023
	Whole body water	−0.56	0.02
	Extracellular water	−1.31	0.016
−ΔBP15 (0-15)	Skiing vs. running	9.32	0.016
	Full vs. half-marathon	8.97	0.009
	Fat-free mass	−0.53	0.228
	Whole body water	−0.71	0.029
	Extracellular water	−1.62	0.028

Parameters: ΔHR5: HR 5 min after finishing—rest time; ΔBP5: BP 5 min after finishing—rest time; −ΔBP15: BP in rest time—15 min after finishing; ΔHR: HR after finishing—after relax time; ΔBP: BP after finishing—after relax time.

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
