# Peer review of "The Impact of Marathons on the Recovery of Heart Rate and Blood Pressure in Non-Professional Male Marathoners’ (≥45 Years)"

_medicina, 2021, doi:10.3390/medicina57121346_

Round 1
Reviewer 1 Report
The present paper aimed at describing the impact of marathon on the recovery of selected physiological parameters in non professionnal athletes aged above 45 years old.
The article in its present form can not be accepted for publication. Here are my suggestions for the revision :
Major :
-Aim of the study : The aim is not clear to me. Do you try to determine physiological parameters involved in the quality of recovery of non professional athletes after a marathon ? or do you try to determine the impact of marathon running on the recovery of BP and HR in these athletes ?
This has to be clarified because the messages are totally differents to the readers.
-Are the authors measured the heart rate variability before and after marathon ? The parameters of HRV can be important to evaluate the potential of recovery of athletes after marathon.
-Also authors did linear regressions between delta BR and delta HR with essentially body composition parameters, it should interesting to test the relationships between other parameters. For instance parameters of HRV with these parameters. Moreover, did the authors measured the sleep qualitity before and after marathon ? This parameter may modulate the recovery too. Relationships between these parameters and echocardiographic (e.g. tissular doppler and skeletal muscle function and soreness parameters could better precise how the recovery is modulated by a marathon running.
These supplementary data are essential to improve the quality of the message of the paper.
-A control group at least, a younger control group has to be added to the study to compare the effet of age on the quality of recovery on non professionnal athletes.
-Marathon is a running event and you included athletes who are training in skiing, running and cycling. How do you take this specific training into account in the results of your study ?
Minor :
-The text has to be proofread by a native English speaker.
-Please rewrite the title to correspond well with the aim of the study.
-The introduction has to focus more on the impact on marathon in non professional runner in one part and on the impact of aging in these athletes in an other part.
-The 2 first paragraph of the introduction have to be shorten.
-In the introduction please explain why you chose athletes >45 yo ?
-In the introduction please expalin why you chose to focus only on BP and HR to explore the impact of marathon on the quality of recovery in non professional athletes aged > 45yo ?
-Please shorten the body composition part of the methods section. You can add a reference and write for example : as previously described, the body composition..."
-Please describe the questionnaire you used.
-Please modify the table 1 with the same format as table 2.
-You did not speak about medication and health condition parameters into the methods section. Why ?
-The figure 2 has to be modified. Please remove the small dotes and only present the 3 types of exercise.
-For the table 3, are there similar results on HR and BP parameters than on there variations between the different times of the study ? Could you also indicates these relationships ?
-Please proofread the paper to correct some mistakes like "seeing" in figure 1 or "HB" line 306.
-Please could you explain what did you want to say with the word "taxing" line 307 ?
-The discussion and the conclusion sections have to be totally rewritten and focused on the results of the present study.
Author Response
Dear reviewer, here are our comments.
Major :
-Aim of the study : The aim is not clear to me. Do you try to determine physiological parameters involved in the quality of recovery of non professional athletes after a marathon ? or do you try to determine the impact of marathon running on the recovery of BP and HR in these athletes ? This has to be clarified because the messages are totally differents to the readers.
Thank you for your excellent remark. Our aim was to determine the impact of marathon and cofactors associated with marathons on the recovery of heart rate (HR) and blood pressure (BP) of non-professional male athletes aged above 45 years old who are at risk based on the literature (this is also shown by the linear regression analysis). The aim of the research is also specified in the manuscript: “to describe the impact of marathon on the HR and BP recovery of selected physiological parameters in non-professional male athletes aged above 45 years old who are at risk based on the literature”.
-Are the authors measured the heart rate variability before and after marathon? The parameters of HRV can be important to evaluate the potential of recovery of athletes after marathon.
HR and BP were determined at rest 24-48 hours before the start of the marathon. With such a large number of subjects at the competition site just before the start, it would not have been possible to determine these parameters in any way. The timing of the determination of HR and BP at rest was also specified in the manuscript (Materials and Methods). Post-marathon measurements are described in detail in the methodology chapter also.
-Also authors did linear regressions between delta BP and delta HR with essentially body composition parameters, it should interesting to test the relationships between other parameters. For instance parameters of HRV with these parameters.
Besides to assess the impact of different marathons to the HR and BP recovery the different cofactors (not only associated with body composition) factors as follows: age, incidence of diagnosed chronic disease (separately CV); previous participation in different marathons and passed km-s; preparation for marathon (hours and times per week) during 3 months, consumption of liquids 1 day prior and during marathon; whole and extracellular liquid; were linked in linear regression analyses (statistic section). If the p value vas <0.05 we did not show it on the table. The results of correlation between some parameters is showed in the text; mostly in paragraph 3.2 (for example body mass was not associated with previously passed km-s, and so)…
Moreover, did the authors measured the sleep qualitity before and after marathon?
Unfortunately, we did not study the quality of sleep. A corresponding note is also added at the end of the manuscript (limitations of the study).
This parameter may modulate the recovery too. Relationships between these parameters and echocardiographic (e.g. tissular doppler and skeletal muscle function and soreness parameters could better precise how the recovery is modulated by a marathon running. These supplementary data are essential to improve the quality of the message of the paper.
Thank you for your good comment, but unfortunately we did not specify these parameters. Determining such parameters at the finish is extremely difficult. It must be taken into account that after completing the marathon, the finishers want to rest, wash themselves and eat. A dozen subjects can also finish the marathon at the same time, and it is impossible for them to perform such analyzes at the same time due to the abundance of equipment and space resources (for example, there are practically no buildings at the finish of the ski marathon).
-A control group at least, a younger control group has to be added to the study to compare the effet of age on the quality of recovery on non professionnal athletes.
Thank you for your excellent remark. The aim of the study was not to compare different age groups, but clarify, does the age impacts and how the recovery of HR and BP inside the risk group (men over the age of forty-five).
-Marathon is a running event and you included athletes who are training in skiing, running and cycling. How do you take this specific training into account in the results of your study?
Thank you for your excellent remark. If in the past the concept of a marathon was primarily related to running, today we also see marathon events in other sports. Our goal was to assess the extent to which men are preparing for different marathons, ie training. We have heard, that same of our marathon participants does not prepare for marathon or do it very few hours and times before marathon. We assumed that, in fact, the subjects are rather inactive in their free time. So we ask all activities they do prior marathon. Certainly, different training methods have different effects and this can affect the results, but the aspects studied provide an overview of the training habits of non-professional enthusiasts participating in the marathon. The activities prior marathon is presented in paragraph 3.1. We can say that our participants were very active, unfortunately we cannot answer, was this activity adequate. We also monitored participants’ activity bound up with their job.
However, in this paragraph the sentence “The preparation style was proportional with participators in different marathons” was added.
Minor:
-The text has to be proofread by a native English speaker.
Answer: This manuscript has undergone English language editing by MDPI. The text has been checked for correct use of grammar and common technical terms, and edited to a level suitable for reporting research in a scholarly journal. MDPI uses experienced, native English speaking editors.
-Please rewrite the title to correspond well with the aim of the study.
Thank you for your excellent remark, the title has been reworded to reflect the stated purpose: “The Impact of Marathon on the Recovery of Heart Rate and Blood Pressure in Non-Professional Male Marathoners (≥45 years)”.
-The introduction has to focus more on the impact on marathon in non professional runner in one part and on the impact of aging in these athletes in an other part.
Thank you for this comment. The introduction section has been significantly shortened. However, we also consider it important to emphasize age-related issues, as the risk of training increases with age.
-The 2 first paragraph of the introduction have to be shorten.
Thank you for your excellent remark, the part of Introduction is significantly shortened.
-In the introduction please explain why you chose athletes >45 yo ?
A description of the choice of subjects is given in the literature review, e.g. “over 45-year-old men belong to a risk group [16,20]” (line 80). We also added a short clarification to the beginning of page 3.
-In the introduction please expalin why you chose to focus only on BP and HR to explore the impact of marathon on the quality of recovery in non professional athletes aged > 45yo ?
Thank you for your good comment. Our choice in favor of BP and HR is due to the fact that determining other parameters at the finish is extremely difficult. It must be taken into account that after completing the marathon, the finishers want to rest, wash themselves and eat. A dozen subjects can also finish the marathon at the same time, and it is impossible for them to perform such analyzes at the same time due to the abundance of equipment and space resources (for example, there are practically no buildings at the finish of the ski marathon).
-Please shorten the body composition part of the methods section. You can add a reference and write for example : as previously described, the body composition..."
Thank you for your excellent remark, the part of Methods (body composition) is significantly shortened.
-Please describe the questionnaire you used.
The section of the questionnaire is described in the methodology chapter as follows: “Firstly, 24–48 hours prior to the stage, the study participants filled out a questionnaire about their age, incidence of previously diagnosed chronic disease, previous participation in different marathons and passed kilometers, preparation for marathon (hours and times per week) during the prior 3 months, and their eating habits. The questionnaire was composed by the authors of the article, and tested (pilot study) using a representative sample including 8 physically active adult men. As a result of the pilot study, the description of sports equipment and training conditions in the questionnaire was shortened; the terms used in the questionnaire were adjusted to make them more comprehensible to the respondent.”
-Please modify the table 1 with the same format as table 2.
Unfortunately we cannot understand the comment.
-You did not speak about medication and health condition parameters into the methods section. Why ?
Thank You. We had described in methods shortly our questionnaire, but we had forgot from there question about medication. Now it is added: We Firstly, 24–48 hours prior to the stage, the study participants filled out a questionnaire about their age, incidence of previously diagnosed chronic disease, used medications, previous participation in different marathons.
-The figure 2 has to be modified. Please remove the small dotes and only present the 3 types of exercise.
Thank you for your excellent remark, the dotes are removed and only the 3 types of exercises presented. We added in figure text: Figure 2. Three mean activities during three-month preparation for marathon (in numbers presented times and hours preparing for marathon independent activities, fitness studios, gymnasium, swimming and garden tools are included).
-For the table 3, are there similar results on HR and BP parameters than on there variations between the different times of the study ? Could you also indicates these relationships ?
Thank you, this is changed now.
After relax time the recovery was supposed and none of monitored factors were influence HR recovery. Monitoring BP recovery, the skiers’ vs runners and full vs half marathoners the results differences where higher (likely due to results after finishing) in comparing changes during relax time. As in recovery time in subjects with higher fat free mass and with greater amount of whole body and extracellular liquid was smaller, after recovery time the results of these persons were better.
-Please proofread the paper to correct some mistakes like "seeing" in figure 1 or "HB" line 306.
Thank you for pointing out, typos have been fixed.
-Please could you explain what did you want to say with the word "taxing" line 307 ?
The proofreaders suggested "taxing" as a synonym for "burdensome".
-The discussion and the conclusion sections have to be totally rewritten and focused on the results of the present study.
Thank you for your excellent remark, the discussion chapter has been significantly shortened and irrelevant deleted.
Thank you and best wishes
Reviewer 2 Report
This study aims to examine certain factors influencing non-professional older male marathoners’ heart rate and blood pressure, and their recovery. The study is quite intriguing with the appropriate and thorough methods. Statistical analysis was conducted in a very precise manner, with results well presented. However overall writing of the manuscript can be significantly improved. Please see some of the comments and raised issues below.
General comments:
Introduction:
- First paragraph is too general. We all know general benefits of the exercise. I believe that authors should focus directly on how all endurance sports influences health (e.g. half-marathons, marathons, ultra-marathons, long distance swimming, skiing and cycling...).
- Introduction is quite comprehensive. Authors should be more concise, rather than trying to say everything. Please consider shortenings the introduction so it can be more appealing to the potential readers.
- While reading the introduction I’ve noticed that the focus was primarily on the marathon running. However, only 26 out of 136 participants are marathoners. Introduction needs to cover more of long distance skiing and cycling, and not just long distance running. Consider some new references, such as: Rauter, S. (2014). Mass sports events as a way of life (differences between the participants in a cycling and a running event). Kinesiologia Slovenica, 20(1).
Methods:
- Methods are written thoroughly, however, the part of body impedance measurements is written in too much detail. Please consider deleting it and just cite adequate paper explaining this method.
- Authors should clearly present the exact distances for ski and cycling events. I failed to notice that within the manuscript (only in Figure 1). Besides that, comparing long distance running, skiing and cycling is not quite adequate. Musculoskeletal work, pacing, cardiovascular work, everything differs a lot. Further explanations from authors are needed regarding this issue as well as a clear rationale why to include all this events.
Results:
- I believe that pacing in marathon can cause either higher or lower HR and BP after the race. Please consider explaining how pacing (in particular slowing down in the last 10-15km) can influence HR and BP. Moreover, men are more likely to slow down than women. Regarding that, please consider this reference: Cuk I, Nikolaidis P., Knechtle B., Sex differences in pacing during half-marathon and marathon race, Research in Sports Medicine, Vol. 28, No. 1, pp. 111 - 120, Jan, 2020.
Discussion:
- Discussion is just too extensive and hard to read. I felt that authors wanted to elaborate everything in small detail, which is sometimes unnecessary. I don’t have any specific comment regarding the discussion. Just try to make it shorter and more appealing to the readers.
Specific comments:
Page 2, Lines 46 to 51: Please include this novel study as a reference regarding the participation in the long distance events: “Nikolaidis P., Cuk. I, Suárez V., Villiger E., Knechtle B., Number of finishers and performance of age group women and men in long distance running: comparison among 10km, half-marathon and marathon races in Oslo, Research in Sports Medicine, Vol. 29, No. 1, pp. 55 - 66, 2021”.
Page 3, Lines 110-111: Please provide a reference to this pilot study, or present some results of the validity and reliability of this questionnaire.
Page 8, Line 155:
Figure 1: Please increase resolution of this figure.
Author Response
Dear reviewer, here are our comments!
General comments:
Introduction:
- First paragraph is too general. We all know general benefits of the exercise. I believe that authors should focus directly on how all endurance sports influences health (e.g. half-marathons, marathons, ultra-marathons, long distance swimming, skiing and cycling...).
Thank you for your excellent remark. The introductory part has been significantly shortened and text not directly related to the topic has been deleted.
- Introduction is quite comprehensive. Authors should be more concise, rather than trying to say everything. Please consider shortenings the introduction so it can be more appealing to the potential readers.
Thank you for your excellent remark. The introductory part has been significantly shortened and text not directly related to the topic has been deleted.
- While reading the introduction I’ve noticed that the focus was primarily on the marathon running. However, only 26 out of 136 participants are marathoners. Introduction needs to cover more of long distance skiing and cycling, and not just long distance running. Consider some new references, such as: Rauter, S. (2014). Mass sports events as a way of life (differences between the participants in a cycling and a running event). Kinesiologia Slovenica, 20(1).
Thank you for this excellent suggestion. We added the publication to the manuscript.
Methods:
- Methods are written thoroughly, however, the part of body impedance measurements is written in too much detail. Please consider deleting it and just cite adequate paper explaining this method.
Thank you for your excellent remark, the part of methodology has been significantly shortened.
- Authors should clearly present the exact distances for ski and cycling events. I failed to notice that within the manuscript (only in Figure 1). Thank you for the comment, the information regarding the distances is added to the text. Besides that, comparing long distance running, skiing and cycling is not quite adequate. Musculoskeletal work, pacing, cardiovascular work, everything differs a lot. Further explanations from authors are needed regarding this issue as well as a clear rationale why to include all this events.
Thank you, the explanation is added to the text.
Results:
- I believe that pacing in marathon can cause either higher or lower HR and BP after the race. Please consider explaining how pacing (in particular slowing down in the last 10-15km) can influence HR and BP. Moreover, men are more likely to slow down than women. Regarding that, please consider this reference: Cuk I, Nikolaidis P., Knechtle B., Sex differences in pacing during half-marathon and marathon race, Research in Sports Medicine, Vol. 28, No. 1, pp. 111 - 120, Jan, 2020.
Thank you for this suggestion, we looked recent articles by Nikolaidis and considered to add article (Nikolaidis et al. 2019) to the discussion chapter. We already have used the ideas from Cuk et al. 2019 (co-author to Nikolaidis). We significantly shortened the introduction chapter and decided to delete the gender differences due to the size of the chapter.
Discussion:
- Discussion is just too extensive and hard to read. I felt that authors wanted to elaborate everything in small detail, which is sometimes unnecessary. I don’t have any specific comment regarding the discussion. Just try to make it shorter and more appealing to the readers.
Thank you for your excellent remark, the discussion chapter has been significantly shortened and irrelevant deleted.
Specific comments:
Page 2, Lines 46 to 51: Please include this novel study as a reference regarding the participation in the long distance events: “Nikolaidis P., Cuk. I, Suárez V., Villiger E., Knechtle B., Number of finishers and performance of age group women and men in long distance running: comparison among 10km, half-marathon and marathon races in Oslo, Research in Sports Medicine, Vol. 29, No. 1, pp. 55 - 66, 2021”.
Thank you for this suggestion, we looked at other recent articles by Nikolaidis and considered it important to add the article to the Introduction.
Page 3, Lines 110-111: Please provide a reference to this pilot study, or present some results of the validity and reliability of this questionnaire.
We have not published the results of the pilot study, and this is not because it is only part of the process of creating the questionnaire, we have not considered it necessary.
Page 8, Line 155:
Unfortunately, we do not understand this comment.
Figure 1: Please increase resolution of this figure.
Thank you for this comment, the resolution is fixed.
Best wishes!
Round 2
Reviewer 1 Report
Thank you for your answers. The article is better in the present form.
Reviewer 2 Report
Thank you for adequately answering to all of my questions and raised issues.
This manuscript is a resubmission of an earlier submission. The following is a list of the peer review reports and author responses from that submission.